# Hypergraph-guided Intra- and Inter-category Relation Modeling for Fine-grained Visual Recognition

## ABSTRACT

Fine-grained Visual Recognition (FGVR) aims to distinguish objects within similar subcategories. Humans adeptly perform this challenging task by leveraging both intra-category distinctiveness and inter-category similarity. However, previous methods fail to combine these two complementary dimensions and mine the intrinsic interrelationship among various semantic features. To address the above limitations, we propose HI2R, a Hypergraph-guided Intra- and Inter-category Relation Modeling approach, which simultaneously extracts the intra-category structural information and inter-category relation information for more precise reasoning. Specifically, we exploit a Hypergraph-guided Structure Learning (HSL) module, which employs hypergraphs to capture high-order structural relations, transcending traditional graph-based methods that are limited to pairwise linkages. This advancement allows the model to adapt to significant intra-category variations. Additionally, we propose an Inter-category Relation Perception (IRP) module to improve feature discrimination across categories by extracting and analyzing semantic relations among them. Our objective is to alleviate the robustness issue associated with exclusive reliance on intra-category discriminative features. Furthermore, a random semantic consistency (RSC) loss is introduced to direct the model's attention to commonly overlooked yet distinctive regions, indirectly enhancing the representation ability of both HSL and IRP modules. Both qualitative and quantitative results demonstrate the effectiveness and usefulness of our proposed HI2R model.

## CCS CONCEPTS

• **Computing methodologies → Object recognition**.

## KEYWORDS

Fine-grained Visual Recognition, Hypergraph, Vision Transformer, Intra- and Inter-category, Random Semantic Consistency

## 1 INTRODUCTION

Fine-grained visual recognition (FGVR) [39] has been a fundamental yet challenging task in computer vision, intending to distinguish subordinate categories within a broader category. Unlike general object recognition, FGVR faces increased difficulty due to large intra-category variations and small inter-category differences. These challenges are further compounded when objects from different

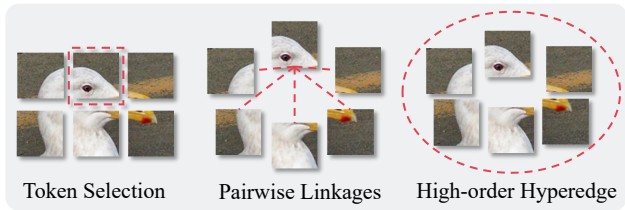

**(a) Intra-category feature learning in different methods**

Token Selection    Pairwise Linkages    High-order Hyperedge

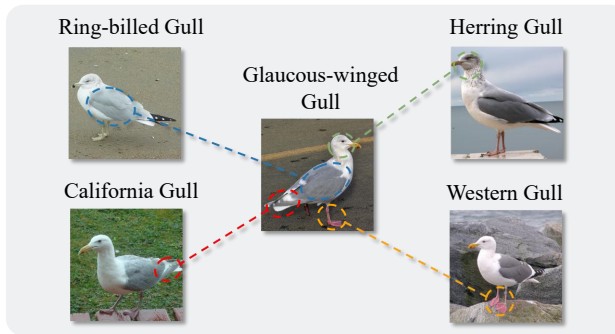

Ring-billed Gull    Herring Gull

Glaucous-winged Gull

California Gull    Western Gull

**(b) Inter-category relation perception among different categories**

**Figure 1: The illustration of some limitations in previous FGVR works. (a) The comparison of ViT, graph [30] and hypergraph in intra-category feature learning, taking the bird head as a representative example. (b) Challenges in direct category determination through a single local feature.**

categories share similar textural characteristics and appear in the same visual context, making it difficult to differentiate local features. Furthermore, even within a single category, significant variation in appearance, shape, and pose complicates the recognition process.

Preceding works in FGVR have successfully employed attention mechanisms in CNNs to highlight and utilize salient regions [8, 9, 27, 28, 49, 50]. However, the conventional CNN-based approach of enhancing receptive fields through layer stacking, while capturing more contextual information, inadvertently blurs fine-grained details, compromising both robustness and discriminative ability in subcategory recognition. In response, recent research has shifted towards capturing intricate local features to deal with the challenge of large intra-category variations. Vision Transformers (ViT) [10] and its variants leverage the self-attention mechanism that naturally emphasizes local features by processing images in a patch-by-patch manner. This allows ViT to capture fine-grained details, effectively overcoming the limitations of traditional CNNs in FGVR tasks [19, 21, 35, 40, 45, 46]. Despite these advancements, many methods still primarily concentrate on significant patch token selection (see in Fig.1a), while overlooking the intricate high-order

relations among various local semantic features. Recognizing these structural relations is crucial for accurately identifying object foreground parts, which further helps reduce background distractions and enhances classification stability [5, 30]. For example, distinguishing a bird's gray legs from dense foliage requires a deep understanding of the structural relations among different bird parts.

To address these challenges, some research has explored the utilization of graphs to capture relational information [30, 36, 48]. Although graph structures provide flexibility and effectiveness in relation modeling, a fundamental limitation [18, 23] is their restriction to exclusively connecting two nodes, accommodating only pairwise relations. This is inadequate for fine-grained objects whose parts (like a bird's head in Fig. 1a) may be represented across multiple patch tokens with complex dependencies. Confining these complex multivariate relations to binary connections may lead to redundant calculations or the loss of crucial information [2]. Inspired by this, our research focuses on effectively modeling complex higher-order structural relations to better discern intra-category differences.

In addition, for the challenge of small inter-category differences, previous works mainly guide the model to extract more discriminative features [3]. Nevertheless, these approaches primarily focus on extracting global features or those at a single level of granularity. The limited and uniform nature of the information encoded in these features often leads to pixel misidentification. For example, as illustrated in Fig.1b, discerning the category of the central image based solely on one single local feature is challenging due to the similarity of its local features to those in other categories. Therefore, another focus of this paper is on leveraging inter-category relations to provide supplementary guidance for reasoning, tackling the challenge of small inter-category differences.

Considering the above factors, we propose a Hypergraph-guided Intra- and Inter-category Relation Modeling **(HI2R)** for FGVR. It utilizes both intra-category high-order relations and inter-category similarity relations to enhance the discriminative power of feature representations. **Firstly**, to cope with significant intra-category variance, we introduce a hypergraph-guided structural learning (HSL) module that constructs patch tokens into a hypergraph. This structured modeling approach leverages hypergraphs to connect an arbitrary number of nodes within one hyperedge, thus facilitating the effective representation of complex spatial relations among semantic features. Additionally, we implement spatial background suppression to reduce noise introduced into the hypergraph. **Secondly**, the Inter-category Relation Perception (IRP) module extracts discriminative semantic relation information across categories to address the challenge posed by small inter-category differences. By modeling the similarity distribution within each category, this module facilitates a more nuanced understanding of inter-category relations. It enhances the original discriminative features through integrating cross-category shared semantics, thereby improving the model's ability to distinguish among similar categories. **Finally**, to tackle the potential semantic alignment challenges within the backbone network ViT [15] and further bolster the representational capabilities of the HSL and IRP modules, we introduce the Random Semantic Consistency (RSC) loss. Visualization experiments clearly demonstrate (Fig. 5) that this loss formulation encourages the model to delve deeper into diverse fine-grained representations, further

improving the robustness of the overall system against various distortions and perturbations.

Our main contributions are summarized as follows:

- We propose a Hypergraph-guided Intra- and Inter-category Relation Modeling method, which emphasizes both intra-category structural features and inter-category semantic features. By collaboratively leveraging these two complementary dimensions, our method cooperates to promote model performance improvement.
- To deal with the challenge of large intra-category differences, we propose a hypergraph-guided structural learning module. It extracts intrinsic higher-order structural attributes that are robust to intra-category differences. To our knowledge, this marks the first application and validation of hypergraph structures in FGVR.
- To further enhance discrimination across different categories, we propose a novel inter-category relation perception module that extends inter-category variability. Additionally, a random semantic consistency loss is introduced to guide the model's focus towards commonly neglected yet distinctive areas.
- Extensive experimental results validate the effectiveness of our proposed methods, outperforming current state-of-the-art approaches across diverse test settings.

## 2 RELATED WORK

### 2.1 Vision Transformer in FGVR

By virtue of its patch-based processing and powerful self-attention mechanism, ViT [10] exhibits superior capacities over traditional CNNs, resulting in enhanced performance. Following this advancement, several methods [19, 21, 35, 40, 45, 46, 51] have been proposed to improve the performance of FGVR. Among these, RAMS-Trans [21] proposes a recursive approach leveraging self-attention weights to learn discriminative parts in a multi-scale way. To capture subtle visual differences between different subordinate categories in ViT, DCAL [51] introduces a dual cross-attention learning algorithm to complement self-attention learning for capturing visual differences. Meanwhile, MpT-Trans [35] extends the class token to multiple tokens representing various parts, improving the extraction of discriminative information. However, while these methods offer significant benefits, they primarily focus on learning discriminative features without considering the relations among various semantic features. To bridge this gap, our proposed HI2R innovatively leverages rich semantic relations across different categories, utilizing the IRP module for enhanced reasoning.

### 2.2 Graph/Hypergraph Structure Learning

Graph neural networks (GNNs) have gained popularity for their ability to represent unstructured data as graphs, with diverse applications ranging from social networks [17] to recommendation systems [37], and other fields. This versatility extends to FGVR, where recent efforts have explored graph structures to represent image elements, enhancing the analysis of complex visual data. For instance, GaRD [48] introduces a graph grouping module to map features onto a low-dimensional manifold, preserving rich

**Figure 2: Overview of our proposed HI2R model. Fine-grained images and their transformed counterparts are fed into the backbone network. The $\mathcal{L}_{RSC}$ is employed for enforcing the attention consistency. Additionally, patch tokens are processed through the HSL module to construct the hypergraph. Following this, the structural features derived from the HSL module are concatenated with the class token to generate the fusion feature. This fusion feature is then expertly utilized by the IRP module to discern inter-category relations, thereby enriching the model's analytical capabilities. The $\mathcal{L}_{CE}$ represents classification loss.**

semantic relationships for fine-grained recognition. SIM-Trans [30] constructs a graph to represent object structures, which are then extracted and integrated into the backbone. Additionally, SRGN [36] deduces structure embedding by correlating position information with visual features along the axial direction. Despite these advancements, the traditional graph structure, primarily reliant on pairwise connections, faces limitations in expressing multifaceted relationships inherent in complex data.

To overcome this, hypergraphs are employed for high-order feature representation, with Hypergraph neural networks (HGNNs) extending the capabilities of GNNs to encompass more intricate data structures. DHGNN [23] employs K-Means and KNN algorithms to construct a dynamic hypergraph structure, enhancing data representation precision. Similarly, ViHGG [18] innovates by establishing and updating the hypergraph structure with the Fuzzy C-Means method, improving computational efficiency. Inspired by ViHGG [18], HI2R utilizes hypergraphs to capture high-order structural relations among semantic features. To specifically cater to FGVR challenges, we introduce a spatial-based background suppression operation to mitigate the impact of background noise, thereby enhancing discriminative representation.

## 3 APPROACH

As illustrated in Fig.2, our proposed HI2R incorporates the HSL module, which processes patch tokens derived from the $L$-th transformer

layer and leverages hypergraph structures to effectively capture spatial information (§3.2). Additionally, the IRP module employs cross-attention mechanisms to extract and utilize inter-category semantic relations, enhancing reasoning through additional guidance (§3.3). Furthermore, the integration of RSC loss facilitates the learning of diverse discriminative features, thereby indirectly boosting the representational power of both the HSL and IRP modules (§3.4).

### 3.1 The Backbone Network

Following previous methods, ViT is utilized as the backbone network for feature extraction. This section briefly revisits the key components of ViT, including tokenization, position embeddings, and encoder blocks, which play a crucial role in this research.

An image $\mathcal{I}$ is first split into $N = H \times W$ patches $x^i \in \mathbb{R}^{P \times P \times C}$, $i \in \{1, \ldots, N\}$, where $P$ is the size of each patch, $C$ is the channel number of the image, and $N$ is the number of patches. Subsequently, a linear embedding layer $E \in \mathbb{R}^{(P^2 \cdot C) \times D}$ is employed to map each patch into a token, with $D$ being the dimension of tokens. In addition, a learnable class token $x_{cls} \in \mathbb{R}^D$ is introduced for classification and a trainable positional encoding $E_{pos} \in \mathbb{R}^{(N+1) \times D}$ is incorporated. Consequently, the input to the first transformer layer can be represented as follows:

$$z_0 = \left[ x_{cls}; x^1 E; x^2 E; \ldots; x^N E \right] + E_{pos}. \tag{1}$$

There are $L$ transformer layers and each layer is composed of a Multi-Head Self-Attention (MSA) block and a Multi-Layer Perception (MLP) block. Both blocks are preceded by a Layer Normalization (LN) operation. The output of each layer encapsulates the transformations applied by these blocks, and can be summarized as follows:

$$z'_l = MSA\left(LN\left(z_{l-1}\right)\right) + z_{l-1}, \quad i \in (1, \ldots, L),$$
$$z_l = MLP\left(LN\left(z'_l\right)\right) + z'_l, \quad i \in (1, \ldots, L). \quad (2)$$

In the standard ViT model, the class token of the last layer $z_L^{cls}$ is input into a classifier to derive the final classification results.

## 3.2 Hypergraph-guided Structure Learning

ViT is adept at encoding region-wise dependencies but tends to neglect high-order structural relations within fine-grained objects. To overcome this limitation, the HSL module is introduced to provide high-order spatial enhancement for further feature representations. However, the direct application of hypergraphs in this context inadvertently introduces background noise. To mitigate this issue, a spatial-based background suppression operation is employed prior to the modeling process.

**Spatial-based Background Suppression**. Constructing a hypergraph directly from all patch tokens obtained from the final transformer layer risks incorporating background noise into our node representations. This noise can detract from the model's performance by obscuring critical features. Consequently, a preliminary step of spatial-based background suppression is proposed to suppress background noise, facilitating the model's ability to learn more meaningful node representations. Assuming there are $N_h$ heads in the MSA block, the attention weights of the $k$-th layer can be represented as follows:

$$a_k^i = \left[a_k^{i_0}, a_k^{i_1}, a_k^{i_2}, \ldots, a_k^{i_N}\right], \quad i \in (1, 2, \ldots, N_h), \quad (3)$$

where $a_k^i$ is the attention weight in the $i$-th head of the $k$-th layer and $a_k^{i_0}$ is the corresponding weight of the class token. Thus, the attention weights of all layers are organized as:

$$A_k = \left[a_k^1, a_k^2, a_k^3, \ldots, a_k^{N_h}\right], \quad k \in (1, 2, ..., L). \quad (4)$$

To further focus attention, we follow attention rollout [1] to integrate attention weights from all preceding layers. Specifically, the aggregated attention is iteratively computed through the matrix multiplication in the following manner:

$$A_{score} = \bar{A}_L \cdot \bar{A}_{L-1} \cdot \cdots \cdot \bar{A}_2 \cdot \bar{A}_1, \quad (5)$$

where $\bar{A} = \frac{1}{2}A_k + \frac{1}{2}E$ illustrates the re-normalized attention weights by using an identity matrix $E$ to account for residual connections, preserving identity information through the layers. Here $A_{score}$ signifies the comprehensive relationships between patch tokens. The weights associated with the class token $a_{score}^{i_0}$ are then extracted from each head in $A_{score}$. In multi-head self-attention, different heads concentrate on distinct aspects. To leverage the unique insights each head offers, weights from various heads are combined. This process can be formalized as follows:

$$F = Top_s\left(\sum_{i=1}^{N_h} a_{score}^{i_0}\right), \quad (6)$$

where $Top_s$ retains the top $s = \lambda \times N$ relevant patch tokens $\left(s \in N^+\right)$, and $F$ represents the selected patch tokens. The term $\lambda$ serves as a hyperparameter that influences the selection ratio. The hypergraph construction is carried out utilizing the subset $F$.

**Hypergraph Construction**. A hypergraph $\mathcal{G} = (\mathcal{V}, \mathcal{E}, \mathcal{W})$ is constructed with hyperedges and nodes, where $\mathcal{V}$ denotes the node set, $\mathcal{E}$ represents the set of hyperedges, and $\mathcal{W}$ signifies the weight matrix of the hyperedge set. Unlike vanilla graphs with pairwise connections, each hyperedge $\mathbf{e}$ can connect an arbitrary number of nodes. In this paper, each patch token derived from spatial-based background suppression is considered a node. The $n$-th node is represented by $f_n \in \mathbb{R}^{1 \times D}$, with all nodes collectively represented as $F = [f_1, f_2, \ldots, f_s] \in \mathbb{R}^{s \times D}$.

To construct the hypergraph structure, the Fuzzy $C$-Means is employed to generate $E$ nodes clusters and a membership matrix $\mathbf{U}$, where each cluster $i$ corresponds to a hyperedge $e_i$ ($i = 1, 2, \ldots, E$). The set of hyperedges is defined as $\mathcal{E} = \{e_1, e_2, \ldots, e_E\}$. Utilizing membership matrix $\mathbf{U}$, the hypergraph $\mathcal{G}$ can be represented by an incidence matrix $\mathbf{H} \in \mathbb{R}^{s \times E}$, where $H_{ij} = U_{ij}$ indicates the membership degree of the corresponding node belonging to the corresponding hyperedge. Following this, the hypergraph convolutional layer then facilitates information exchange between nodes based on the hypergraph $\mathcal{G}$.

**Hypergraph Convolution**. By leveraging the learned $\mathbf{H}$, the model employs a hypergraph convolutional operation to aggregate high-order structural information, thereby enhancing feature representations. This operation facilitates information exchange among nodes via a two-stage node-hyperedge-node message passing scheme. Initially, it aggregates node features onto hyperedges, followed by aggregating hyperedge features back onto nodes. And the relation-enhanced feature $\mathbf{R} \in \mathbb{R}^{s \times D}$ can be obtained as:

$$\mathbf{R} = \mathbf{D}^{-1/2}\mathbf{H}\sigma\left(\mathbf{W}\mathbf{B}^{-1}\mathbf{H}^T\sigma\left(\mathbf{D}^{-1/2}\mathbf{X}\Theta_1\right)\Theta_2\right), \quad (7)$$

where $\Theta_1$ and $\Theta_2$ are the learnable parameters, and $\sigma$ is an activation function. The multiplication of $\mathbf{H}^T$ facilitates information aggregation from nodes to hyperedges. Conversely, multiplying $\mathbf{H}$ can be considered as information aggregation from hyperedges to nodes. Intermediately, $\mathbf{D} \in \mathbb{R}^{s \times s}$ and $\mathbf{B} \in \mathbb{R}^{E \times E}$ denote the node degree matrix and the hyperedge degree matrix obtained through the broadcast operation, respectively. To further enhance the feature transformation capacity and mitigate the over-smoothing phenomenon, a Multilayer Perceptron (MLP) is applied to each node:

$$Z = \sigma\left(\mathbf{R} \cdot W_1\right) \cdot W_2 + \mathbf{R}, \quad (8)$$

where $Z \in \mathbb{R}^{s \times D}$, $W_1$ and $W_2$ are the weights of fully-connected layers, with the bias term omitted for simplicity. In both the Hypergraph Convolution and MLP modules, batch normalization is applied following each fully connected and hypergraph convolution layer, which is not shown for brevity.

The updated feature embeddings $Z$ are considered as the structural feature, which is subsequently flattened and concatenated with (represented in Fig.2 as ©) the class token $z_L^{cls}$ to form an aggregated feature. This feature introduces structural information into the transformer backbone. The aggregated feature is then input into a fully connected layer for feature fusion:

$$\mathbf{X} = RELU\left(\left[\varphi\left(\mathbf{Z}\right); z_L^{cls}\right] \cdot W_3\right) + b, \quad (9)$$

where $\varphi$ denotes the flatten operation, $W_3$ and $b$ represent the weight and bias in the fully connected layer, respectively. And $\mathbf{X}$ signifies the fusion feature. Through end-to-end training, the model effectively understands the composition of fine-grained objects and places emphasis on spatial-based feature embeddings. Consequently, the transformer network is equipped to capture both the appearance and structural information of the objects, facilitating accurate recognition.

## 3.3 Inter-category Relation Perception

Relying solely on intra-category discriminative features without considering semantic relations across categories may lead models to overemphasize specific details. This tendency often results in confusion, especially when processing images with similar local features. Therefore, it is equally essential to focus on inter-category semantic relations, which improves model robustness by discerning subtle differences across categories. The proposed IRP module capitalizes on these relations to refine the fusion feature, thereby boosting overall accuracy and performance. Specifically, a concept cache is maintained, denoted by $\mathbf{P} = \{p_c \mid i = 1, 2, \ldots, C\}$, where $C$ represents the count of categories in the dataset, and each row $p_c \in \mathbb{R}^D$ denotes a basis concept for the fine-grained category $c$.

**Update of Concept Cache.** To derive the specific concept feature $p_c$ for each instance from category $c$, the fusion feature $\mathbf{X} \in \mathbb{R}^D$, obtained from the HSL module, serves as the extracted concept. Before training, the concept cache $P$ is uniformly initialized. Subsequently, the concept prompt $p_c$ is then dynamically updated using a moving average fashion:

$$p_c \leftarrow (1-\alpha) \cdot p_c + \alpha \cdot \mathbf{X}, \tag{10}$$

where $\alpha \in (0, 1)$ is a coefficient that controls the update rate of the category concept $p_c$.

**Fusion Feature Refinement**. Our goal is to enrich the fusion feature by incorporating knowledge from other categories, thus achieving more informative and comprehensive representations.

Fig.2 illustrates the utilization of a cross-attention mechanism to model affinities among different categories, facilitating the propagation of similarities and distinctions. Finally, we obtain refined feature representations abundant with invariant matching clues. With the fusion feature $\mathbf{X}$ serving as a query and the concept cache $\mathbf{P}$ as both key and value, cross attention refines $\mathbf{X}$ as follows:

$$\mathbf{X}_r = CrossAttention\left(\mathbf{X}, \mathbf{P}\right), \tag{11}$$

where $CrossAttention(\cdot)$ first normalizes the input, then projects $\mathbf{X}$ to $Q \in \mathbb{R}^{1 \times D_h}$, and $\mathbf{P}$ to $K \in \mathbb{R}^{C \times D_h}$ and $V \in \mathbb{R}^{C \times D_h}$. The affinity matrices $A_{cross}$ is computed as:

$$A_{cross} = softmax\left(QK^T / \sqrt{D_h}\right) \in \mathbb{R}^{1 \times C}. \tag{12}$$

The process involves matching $Q$ against $C$ key vectors ($K = [k_1; k_2; ...; k_C]$, $k_i \in \mathbb{R}^{1 \times D_h}$) using inner product, scaled and normalized by softmax. The fusion feature updates with affinity weights, effectuating semantic relation information propagation:

$$\mathbf{X}_r = Q + A_{cross} \cdot V. \tag{13}$$

To some extent, the information propagation procedure can be deduced as a weighted sum of $C$ value vectors ($V = [v_1; v_2; ...; v_C]$,

$v_i \in \mathbb{R}^{1 \times D_h}$). This process maintains moderate computational complexity without introducing additional parameters. Notably, The residual features provide stability in the early training stages.

Following this, the refined feature $\mathbf{X}_r$ is forwarded to a classification head, producing the prediction vector $pred(\mathcal{I})$ for the input image $\mathcal{I}$. Subsequently, the classification loss is defined as:

$$\mathcal{L}_{CE} = - \sum_{\mathcal{I} \in S(\mathcal{I})} \left(y \cdot log\left(pred(\mathcal{I})\right)\right), \tag{14}$$

where $S(\mathcal{I})$ is the training set and $y$ is the one-hot label for $\mathcal{I}$.

## 3.4 Random Semantic Consistency

To further improve the representational capabilities of the HSL and the IRP modules, the RSC loss is introduced. This loss encourages the model to delve deeper into diverse label-relevant regions. As shown in Fig.2, the architecture incorporates a dual-branch module, each starting with a ViT backbone and culminating in a classification header. The parameters of the two branches are shared.

Giving a batch of fine-grained images $\mathcal{I}$, we first randomly erase the input images and then flip these images to get their flipped counterparts $\mathcal{I}' = T(\mathcal{I})$. This mimics large inter-class variations due to varying poses and obscured discriminative features. Each branch processes $\mathcal{I}$ and the transformed image $\mathcal{I}'$ respectively, aggregating attentions as outlined in Eq.5. In the transformer layer, attention weights between patch tokens and the class token indicate their significance in classification. The extracted category-related attention score $A_{score}^{cls} \in \mathbb{R}^{N \times 1}$ is reshaped into $M \in \mathbb{R}^{H \times W}$, where $H$ and $W$ is the number of tokens along image height and width, respectively (See §3.1).

Based on the definition of attention consistency, the attention weights of the transformed image $\mathcal{I}'$ need to be flipped for equivariance. The RSC consistency loss is employed as the disparity between attention weight to minimize the distance between the attention map $M$ and $Flip(M')$:

$$\mathcal{L}_{RSC} = \frac{1}{H \times W} \sum_{i=1}^{H} \sum_{j=1}^{W} \left\| M_{i,j} - Flip(M')_{i,j} \right\|_2, \tag{15}$$

where $\|\cdot\|_2$ represents the Euclidean norm. And $M_{i,j}$ and $Flip(M')_{i,j}$ are the values at the $(i, j)$-th position of the corresponding attention maps $M$ and $M'$, respectively. By minimizing the $\mathcal{L}_{RSC}$ between $M$ and $M'$, the model is encouraged to learn diverse and viewpoint-robust features from a single image.

In summary, the vision transformer backbone and the proposed modules are jointly trained end to end, with the total objective:

$$\mathcal{L} = \mathcal{L}_{CE} + \beta \cdot \mathcal{L}_{RSC}, \tag{16}$$

where $\beta$ is used for the numerical balance of various losses. Training our method only needs the ground-truth label.

## 4 EXPERIMENTS

### 4.1 Datasets and Evaluation Metric

The effectiveness of our proposed HI2R is evaluated through experiments conducted on three benchmarks for fine-grained visual recognition:

- **CUB-200-2011** [34] contains 11,788 images across 200 sub-categories, with 5,994 images for training and 5,794 for testing. It is considered one of the most challenging datasets due to each species having only 30 images for training.
- **NAbirds** [32] is a collection of 48,000 annotated photographs of the 400 species of birds that are commonly observed in North America. Each species is represented by more than 100 photographs, including separate annotations for males, females, and juveniles, resulting in 700 visual categories.
- **iNaturalist2017** [33] is one of the largest FGVR datasets. It features cross-species images with a biased distribution across 5,089 categories. The dataset's large scale and challenging nature provide a robust benchmark for evaluating fine-grained visual recognition performance.

Top-1 accuracy is adopted as the evaluation metric. The model is trained using only image-level labels, without any additional annotations for supervised training.

### 4.2 Implementation Details

Consistent with the settings in previous methods, we emolpy the ViT-Base [10] pre-trained on ImageNet-21k as the backbone network, comprising 12 encoder blocks. The patch size is set to 16, with a stride factor of 12. To ensure a fair comparison with other methods, input images in the CUB-200-2011 and NAbirds datasets are first resized to 600 × 600 and subsequently randomly cropped to 448 × 448 during training. For the iNaturalist 2017 dataset, the image size is set to 304 × 304 following [30]. Random horizontal flipping is applied for training, while center cropping is employed during testing. The model is trained using the stochastic gradient descent (SGD) optimizer with a momentum of 0.9, and is regulated by a cosine annealing scheduler. The initial learning rate is set to 4e-2 for iNaturalist 2017 and 2e-2 for the other two datasets. The model is trained for 10,000 steps, with the first 500 steps as warm-up, and the batch size is set to 16 for all datasets. Therefore, the comparison experiments with state-of-the-art transformer-based methods are conducted under fair and convincing conditions.

In the HSL module, the ratio of selected patch tokens $\lambda$ in spatial-based background suppression and the number of node cluster $E$ for hypergraph construction are set to 0.3 and 6, respectively. In the IRP module, the update rate of category concepts $\alpha$ is set to 1e-2. And $\beta$ is set to 1e-2 when calculating the loss. The experiments are performed using PyTorch on NVIDIA GeForce GTX 3090 Ti GPUs.

### 4.3 Comparison With the State-of-the-Arts

Our proposed HI2R is compared with existing state-of-the-art works, including CNN-based and ViT-based methods on the fine-grained datasets mentioned above.

**CUB-200-2011:** Table 1 illustrates the superior performance of the proposed HI2R over current state-of-the-art approaches. HI2R achieves the best classification accuracy at 92.5%, marking a 1.9% enhancement compared to the ViT baseline [10]. The effectiveness of HI2R is validated through a comparison with various state-of-the-art methods, categorized into two groups. The first group consists of methods using CNNs for feature extraction, with SR-GNN [5] achieving the highest accuracy of 91.9% by leveraging Xception to aggregate context-aware features from relevant image regions.

**Table 1: Comparison experiments with other state-of-theart methods on CUB-200-2011 dataset.**

| Method | Venue | Backbone | Acc(%) |
|---|---|---|---|
| RA-CNN [14] | CVPR 2017 | VGGNet19 | 85.3 |
| MA-CNN [49] | ICCV 2017 | | 86.5 |
| FDL [38] | CVPR 2018 | VGGNet16 | 86.7 |
| HBP [42] | ECCV 2018 | | 87.1 |
| NTS-Net [41] | ECCV 2018 | ResNet50 | 87.5 |
| Cross-X [26] | ICCV 2019 | | 87.7 |
| DCL [6] | CVPR 2019 | | 87.8 |
| PMG [11] | ECCV 2020 | | 89.6 |
| MCEN [25] | ACM MM 2021 | | 89.3 |
| GaRD [48] | CVPR 2021 | | 89.6 |
| CMN [8] | TIP 2022 | | 88.2 |
| MA-ASN [43] | TMM 2022 | | 89.5 |
| SRGN [36] | IJCV 2024 | | 91.4 |
| API-Net [52] | AAAI 2020 | ResNet101 | 88.6 |
| PART [47] | TIP 2021 | | 90.1 |
| CAL [28] | ICCV 2021 | | 90.6 |
| CAP [4] | AAAI 2021 | Xception | 91.8 |
| SR-GNN [5] | TIP 2022 | | 91.9 |
| ViT [10] | ICLR 2020 | ViT-Base | 90.6 |
| RAMS-Trans [21] | ACM MM 2021 | | 91.3 |
| AF-Trans [45] | ICASSP 2022 | | 91.6 |
| TransFG [19] | AAAI 2022 | | 91.7 |
| DCAL [51] | CVPR 2022 | | 92.0 |
| SIM-Trans [30] | ACM MM 2022 | | 91.8 |
| IELT [40] | TMM 2023 | | 91.8 |
| MpT-Trans [35] | ACM MM 2023 | | 92.0 |
| MP-FGVC [24] | AAAI 2024 | | 91.8 |
| **HI2R (Ours)** | - | ViT-Base | **92.5** |

**Table 2: Comparison experiments with other state-of-theart methods on NAbirds dataset.**

| Method | Venue | Backbone | Acc(%) |
|---|---|---|---|
| MaxEnt [13] | NeurIPS 2018 | DenseNet-161 | 83.0 |
| API-Net [52] | AAAI 2020 | | 88.1 |
| DSTL [7] | CVPR 2018 | Inception-v3 | 87.9 |
| Cross-X [26] | ICCV 2019 | ResNet50 | 86.4 |
| PAIRS [16] | WACV 2019 | | 87.9 |
| GaRD [48] | CVPR 2021 | | 88.0 |
| CMN [8] | TIP 2022 | | 87.8 |
| PMG-v2 [12] | TPAMI 2022 | ResNet101 | 88.4 |
| MGE-CNN [44] | ICCV 2019 | | 88.6 |
| FixSENet [31] | NeurIPS 2019 | SENet-154 | 89.2 |
| CAP [4] | AAAI 2021 | Xception | 91.0 |
| SR-GNN [5] | TIP 2022 | | 91.2 |
| ViT [10] | ICLR 2020 | ViT-Base | 89.9 |
| TransFG [19] | AAAI 2022 | | 90.8 |
| IELT [40] | TMM 2023 | | 90.8 |
| MpT-Trans [35] | ACM MM 2023 | | 91.3 |
| MP-FGVC [24] | AAAI 2024 | | 91.0 |
| **HI2R (Ours)** | - | ViT-Base | **91.5** |

Despite the accomplishments of CNN-based methods, ViT-based methods perform better in FGVR. Among ViT-based methods, HI2R demonstrates superior performance, surpassing MpT-Trans [35] by a margin of 0.5%. Furthermore, HI2R exceeds SIM-Trans [30] by an absolute gain of 0.7% in performance. These findings indicate that although the integration of graphs with ViT yields remarkable performance, hypergraphs exhibit an enhanced capability in representing complex high-order structural information in FGVR.

**NABirds:** Table 2 presents the performance of the proposed HI2R in comparison with state-of-the-art methods based on CNNs and ViT. The proposed approach surpasses the ViT baseline [10] by 1.6%. Additionally, it achieves improvements of 0.3% and 0.5% over the leading CNN-based method SR-GNN [5] and ViT-based method MpT-Trans [35], respectively. These methods depended exclusively on the distinctiveness of features within categories for reasoning and neglected the potential benefits of leveraging similarities across categories. This limited and uniform nature of the information encoded in these features further led to pixel misidentification. Our HI2R enhances model performance by concurrently emphasizing intra-category high-order structural features and inter-category discriminative semantic features. Experimental results support the effectiveness of integrating these two complementary dimensions.

**Table 3: Comparison experiments with other state-of-theart methods on iNaturalist2017 dataset.**

| Method | Venue | Backbone | Acc(%) |
|---|---|---|---|
| ResNet152 [20] | CVPR 2016 | ResNet152 | 59.0 |
| SRGN [36] | IJCV 2024 | ResNet50 | 73.6 |
| TASN [50] | CVPR 2019 | | 68.2 |
| SSN [29] | ECCV 2018 | ResNet101 | 65.2 |
| IARG [22] | CVPR 2020 | | 66.8 |
| ViT [10] | ICLR 2020 | | 67.0 |
| RAMS-Trans [21] | ACM MM 2021 | | 68.5 |
| TransFG [19] | AAAI 2022 | ViT-Base | 71.7 |
| AF-Trans [45] | ICASSP 2022 | | 68.9 |
| SIM-Trans [30] | ACM MM 2022 | | 69.9 |
| **HI2R (Ours)** | - | ViT-Base | **73.9** |

**iNaturalist2017:** Table 3 summarizes a comparison between the proposed HI2R and existing state-of-the-art methods. The pure ViT baseline [10] demonstrates a significant advantage over the pure ResNet152 baseline [20], with an absolute improvement of 8.0%, thereby highlighting the effectiveness of the Transformer structure. It is noteworthy that the accuracy of SGRN [36] reaches 73.6%, markedly outperforming all ViT-based methods, which highlights the significance of learning structural information to a certain degree. Nonetheless, compared with SRGN and the optimal ViT-based method TransFG [19], HI2R still realizes additional enhancements of 0.3% and over 2.2%, respectively. This achievement emphasizes the effectiveness of HI2R in leveraging both intra-category structural information and inter-category relation information in FGVR, particularly in the context of large-scale datasets.

## 4.4 Ablation Experiments

We conduct ablation studies to evaluate the contributions of different modules in HI2R and to examine the impact of various settings

**Table 4: Effectiveness of Different Modules in HI2R on CUB-200-2011 dataset.**

| # | Backbone | HSL | IRP | RSC loss | Acc(%) |
|---|---|---|---|---|---|
| 1 | ✓ | | | | 90.60 |
| 2 | ✓ | ✓ | | | 91.42 |
| 3 | ✓ | | ✓ | | 91.21 |
| 4 | ✓ | | | ✓ | 91.16 |
| 5 | ✓ | ✓ | ✓ | | 92.27 |
| 6 | ✓ | ✓ | | ✓ | 92.05 |
| 7 | ✓ | | ✓ | ✓ | 91.78 |
| 8 | ✓ | ✓ | ✓ | ✓ | 92.53 |

on the final classification performance. All ablation experiments are conducted on the CUB-200-2011 dataset, with the observation that similar phenomena can be observed on other datasets as well. The experimental setups are consistent with those described in §4.2.

**Effectiveness of Key Components.** We conduct ablation studies to validate the effectiveness of each key component of the proposed HI2R (including HSL, IRP, and RSC loss). The results are presented in Table 4, and the following conclusions can be drawn:

- Compared to the baseline, HSL, IRP, and RSC loss greatly improves recognition accuracy from 90.60% to 91.42%, 91.21%, and 91.16%, respectively. This demonstrates the effectiveness of each component design in HI2R for FGVR. Specifically, the HSL boosts the model's comprehension of object structure, thereby rendering the feature representation more discriminative. Moreover, the IRP enhances model performance by capturing inter-category semantic relations, thereby providing valuable guidance for reasoning and improving robustness. Furthermore, the RSC loss directs the model's attention towards neglected discriminative areas, consequently enhancing overall model performance.

- The integration of HSL and IRP boosts each other by 0.85% (91.42% vs.92.27%) and 1.06% (91.21% vs.92.27%) on accuracy, respectively, highlighting the crucial role of combining these two complementary dimensions. Furthermore, the introduction of the RSC loss, when combined with HSL, yields an additional accuracy increase of approximately 0.63% (91.42% vs.92.05%). This improvement demonstrates that the RSC loss encourages the model to focus on more diverse features, helping the discovery of additional object areas and thus facilitating intra-object structural learning. Additionally, the combination of the IRP with the RSC loss results in an accuracy improvement of about 0.57% (91.21% vs.91.78%). This boost in performance is attributed to the RSC loss fostering a more comprehensive learning of object features and stabilizing the representation of similarity distributions across categories, which is essential for effective IRP implementation.

- Combining three key components brings an 1.93% improvement in accuracy over the baseline. This indicates the synergistic effect of HSL, IRP, and RSC loss in enhancing intra-category feature distinctiveness and inter-category feature similarity. Such integration facilitates the aggregation of

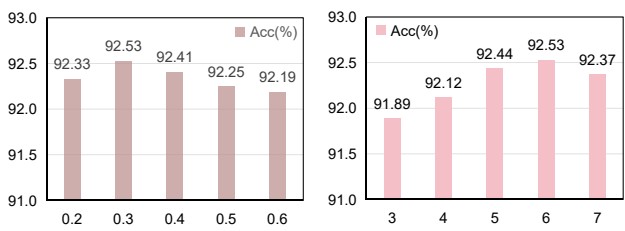

(a) The ratio of selected tokens.  (b) The number of node clusters.

**Figure 3: Evaluations of different $\lambda$ and $E$ in §3.2 on CUB-200-2011 dataset.**

category-related representations and significantly enhances recognition performance.

**Influence of the ratio of selected patch tokens.** We evaluate the impact of adjusting the ratio of selected patch tokens in spatial-based background suppression, which will further influence the hypergraph construction. As illustrated in Fig.3a, setting the ratio to 0.3 yields the highest accuracy performance. Notably, an excessive number of tokens can introduce undesired background noise, negatively affecting classification precision. Conversely, insufficient patch tokens may lead to inadequate representation of local features, thereby reducing model accuracy.

**Influence of the number of node clusters.** We assess the influence of varying node cluster quantities FGVR accuracy. According to Fig.3b, a node cluster count of 6 constitutes the optimal setting for our method. The number of node clusters directly impacts the hypergraph's dimensions by dictating the number of hyperedges. Insufficient hyperedges can lead to reduced performance due to an inadequate representation of data complexities. Conversely, an excess of hyperedges may compromise the model's ability to effectively discriminate local regions, thereby impairing its comprehensive understanding.

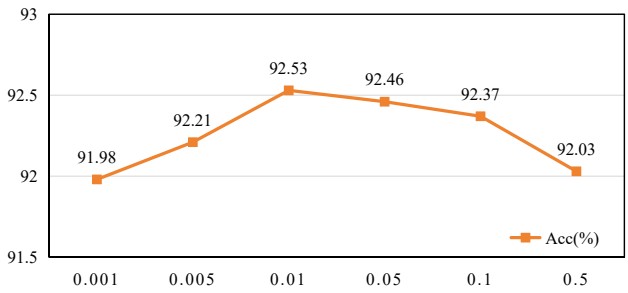

**Figure 4: Evaluation of different update rate of category concepts $\alpha$ in §3.3 on CUB-200-2011 dataset.**

**Influence of the update rate of category concepts.** We analyze the impact of the update rate of category concepts on the performance of the proposed HI2R. As shown in Fig.4, an update rate of 0.01 results in optimal outcomes. In FGVR, it is crucial to distinguish subtle features, requiring rapid adaptation to new samples. A higher update rate for category concepts (larger $\alpha$ values) allows

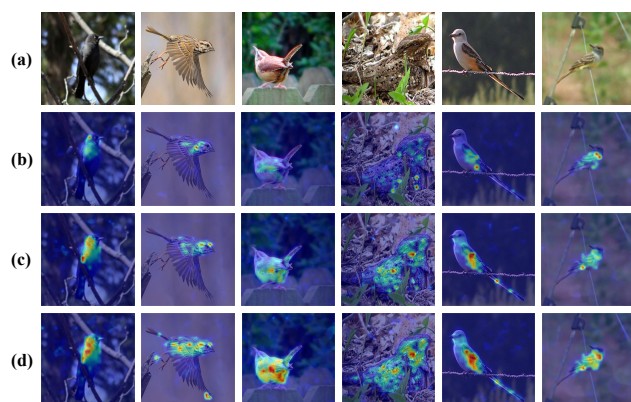

**Figure 5: The visualization of attention maps obtained by different approaches. (a) Raw images. (b) Results of baseline. (c) Results of HI2RD w/o RSC loss. (d) Results of HI2RD.**

the model to rapidly integrate new category features, improving classification accuracy. However, fine-grained datasets are often small, and inter-category differences can be slight. An excessively high update rate may render the model overly sensitive to noise, increasing the risk of overfitting. Therefore, selecting an optimal update rate is essential for balancing rapid adaptation to new samples with the prevention of overfitting. This balance enhances both the accuracy and the stability of the model.

**Visualization.** We present attention maps of different approaches for comparison. As illustrated in Fig.5, compared with ViT which primarily highlights the most discriminative regions of an image, HI2R can more accurately isolate the object foreground by comprehending both intra- and inter-category relations, effectively eliminating irrelevant background details. To demonstrate the influence of the RSC loss, the third and fourth rows of Fig.5 depict the focus regions of HI2R without and with the RSC loss, respectively. These visualizations confirm that the RSC loss directs the network towards identifying a broader range of discriminative features, thus enhancing its effectiveness in recognition.

## 5 CONCLUSION

In this paper, a novel framework has been proposed for FGVR, termed Hypergraph-guided Intra- and Inter-category Relation Modeling (HI2R). This framework is inspired by human visual perception and leverages both intra-category structural relations and inter-category similarity relations to improve FGVR task accuracy. Specifically, the HSL module has employed hypergraphs to encapsulate complex high-order structural relations among local features. Simultaneously, the IRP module has been designed to discern and utilize inter-category relations, serving as additional discriminative signals for more effective reasoning. To mitigate the model's over-reliance on specific semantic features, the RSC loss has been introduced to encourage the exploration of previously neglected discriminative regions, thereby indirectly enhancing the representational quality of both the HSL and the IRP modules. Qualitative and quantitative experiments have consistently demonstrated the superiority of our HI2R framework in the FGVR task.

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
