# OpenReview forum: "Hypergraph-guided Intra- and Inter-category Relation Modeling for Fine-grained Visual Recognition"
_acmmm.org/ACMMM/2024/Conference — MM2024 Poster_

### Official Review · Reviewer_53yb · 2024-05-21

**Rating:** 4
**Confidence:** 3

**Summary:**

Fine-Grained Visual Recognition (FGVR) necessitates distinguishing between similar subcategories, a skill humans excel at by balancing intra-category differences and inter-category similarities.
HI2R innovatively integrates these dimensions using Hypergraph-guided Intra- and Inter-category Relation Modeling, capturing intricate intra-category structures and inter-category relations through Hypergraph-guided Structure Learning (HSL) and Inter-category Relation Perception (IRP) modules.
HSL overcomes pairwise limitations of graph-based methods with high-order hypergraph connections, enhancing intra-category variability understanding, while IRP bolsters cross-category distinction.
Complemented by Random Semantic Consistency (RSC) loss focusing on overlooked yet critical regions, HI2R significantly improves recognition accuracy, as evidenced by empirical results.

**Strengths:**

The author's ideas are simple and easy to understand The writing is clear.

The experiments are relatively sufficient and prove the effectiveness of the method.

**Limitations:**

Several issues require attention:

1. Clarification is needed on the distinction between Concept Cache and category prototypes, as relevant work is currently lacking.

2. Semantic consistency is a common theme in contrastive learning. Could you elaborate on the specifics of the random erasing technique employed? Why were other image augmentation strategies not considered?

3. The paper would benefit from additional comparative studies. For instance:
i) What is the actual gain from background suppression? The spatial background suppression strategy designed by the authors is complex. Could a simpler approach, such as using fully connected layers to predict weights [1], produce comparable performance? Further analysis drawing from related work is missing.

ii) Why was Fuzzy C-Means clustering chosen over other methodologies?

4. The code has yet to be made publicly available.

[1] Progressive Spatio-Temporal Prototype Matching for Text-Video Retrieval

**Suitability:**

2

---

### Official Review · Reviewer_rXHA · 2024-05-24

**Rating:** 5
**Confidence:** 1

**Summary:**

The authors propose HI2R, a Hypergraph-guided Intra- and Inter-category Relation Modeling approach, which simultaneously extracts intra-category structural information and inter-category relation information for more precise reasoning in fine-grained visual recognition. They introduce a Hypergraph-guided Structure Learning (HSL) module to capture high-order structural relations and an Inter-category Relation Perception (IRP) module to improve feature discrimination across categories, addressing limitations of previous methods. Additionally, a random semantic consistency (RSC) loss is implemented to enhance the model's focus on distinctive regions, improving the representation ability of both HSL and IRP modules.

**Strengths:**

1. The motivation of this paper is clear. Leveraging both intra- and inter-category relation for FGVR is interesting and reasonable.
2. This paper proposes two novel module including a hypergraph-guided structural learning module and an inter-category relation perception module.
3. Compared with previous methods, significant performance improvement is observed in this paper.

**Limitations:**

1. Model parameter and computational cost should be provided to for more thorough comparison.
2. In order to prevent background noise, a spatial-based background suppression operation is designed in HSL. What is the performance of the spatial-based background suppression being removed from HSL?
3. What is the connection and difference of the proposed IRP module with prototype learning? I suppose more visualization could be done to show what is learned in the concept cache.

**Suitability:**

2

---

### Official Review · Reviewer_6LXt · 2024-05-25

**Rating:** 4
**Confidence:** 3

**Summary:**

This paper addresses Fine-grained Visual Recognition (FGVR). The authors propose HI2R, a Hypergraph-guided Intra- and Inter-category Relation Modeling approach, to combine intra-category distinctiveness and inter-category similarity effectively. HI2R includes a Hypergraph-guided Structure Learning (HSL) module to capture high-order structural relations and an Inter-category Relation Perception (IRP) module to improve feature discrimination across categories. Additionally, a random semantic consistency (RSC) loss is introduced to enhance the model's attention to distinctive regions. The experiments demonstrate the superior performance of the proposed model.

**Strengths:**

The paper is well-written and easy to follow.

All the modules of the model proposed by the author have proved to be effective.

**Limitations:**

1 The application of GNN technology appears to be a common practice. For example, [1] and [2] have applied this technology. The authors have also discussed related methods in the related work section. However, it is necessary to compare this work with those methods, especially the relevant ViHGG, and point out the differences. This comparison is crucial for assessing the contributions of this paper.

[1] Region Graph Embedding Network for Zero-Shot Learning. ECCV 2020.

[2] Dual Part Discovery Network for Zero-Shot Learning. MM 2022.

2 The impact of $\beta$ has not been studied.

3 In some places, such as the abstract, "Inter-category Relation Perception" is used, but Figure 2 mentions "Intra-category Relation Perception".

4 The accuracy improvement still seems limited.

Overall, the modules designed in this paper are effective, and the motivation is clear. However, some issues still need further attention.

**Suitability:**

2

---

### Official Review · Reviewer_t328 · 2024-05-26

**Rating:** 2
**Confidence:** 4

**Summary:**

This paper proposes a Hypergraph-guided Intra and Inter-category Relation Modeling method for fine-grained visual categorization, which leverages both the intra-category structural relations and inter-category similarity relations for robust feature extraction and classification. Experiments are conducted on three fine-grained datasets.

**Strengths:**

1.	The proposed method is sound and the procedures are easy to understand.
2.	This paper is well-written and easy to follow.

**Limitations:**

1.	The motivation of the proposed method should be further clarified. Why does the method cluster the significant image patches into different hyperedges in the proposed HSL module and what is its physical meaning? Besides, the high-order hyperedge in Figure 1 should be carefully drawn to emphasize its difference from the previous pairwise linkages.
2.	The novelty should be further clarified. The concept cache in the proposed IRP module seems similar to [a], which keeps the information of different categories. The differences should be highlighted.
3.	The effectiveness of the proposed method should be further validated. In the large-scale fine-grained dataset, i.e., iNaturalist 2017, the proposed HI2R method only surpasses the sub-optimal SRGN [b] by 0.3% with a stronger ViT backbone. More explanations should be added.
4.	More visualization results should be presented to show the image patches in the proposed hypergraph for readability.
[a] Fine-Grained Classification via Categorical Memory Networks. TIP 2022
[b] Accurate fine-grained object recognition with structure-driven relation graph networks. IJCV 2024

**Suitability:**

2

---

### Meta-Review · Area_Chair_yaGV · 2024-06-29

**Recommendation:** Accept (Poster)
**Confidence:** 3

**Metareview:**

The reviewers have some concerns on the motivations, novelties, experimental validation, and computational cost of the proposed method. After rebuttal, three reviewers agree to accept this paper while one reviewer insists of weakly rejecting the paper. Though the performance gain of the proposed method on some datasets is limited, the paper is well written with clear motiviations, and the novel components are shown to be effective in experiments. I hence decide to accept this paper.